# Desert-adapted tomato *Solanum pennellii* exhibit unique regulatory elements and stress-ready transcriptome patterns to drought

J. Sebastián Contreras-Riquelme[1,2,3☉], Miguel Contreras[1,2,3☉], Tomas C. Moyano[1,2,3],
Rachid Sjoberg [1,2,3], José Jimenez-Gomez [4], José M. Alvarez [1,2,3]*

1 Centro de Biotecnología Vegetal, Facultad de Ciencias de la Vida, Universidad Andrés Bello, 2 ANID Millenium Institute for Integrative Biology iBio, Chile, 3 ANID Millenium Nucleus in Data Science for Plant Resilience (Phytolearning, 4 Centro de Biotecnología y Genómica de Plantas (CBGP, UPM-INIA) Universidad Politécnica de Madrid (UPM)—Instituto Nacional de Investigación y Tecnología Agraria y Alimentaria (INIA, CSIC), Campus de Montegancedo, Pozuelo de Alarcón, Madrid, Spain

☉ These authors contributed equally to this work.
* jose.alvarez.h@unab.cl (JMA)

## Abstract

Drought is a significant environmental stressor that severely impairs plant growth and agricultural productivity. Unraveling the molecular mechanisms underlying plant responses to drought is crucial for developing crops with enhanced resilience. In this study, we investigated the transcriptomic responses of cultivated tomato (*Solanum lycopersicum*) and its drought-tolerant wild relative, *Solanum pennellii*, to identify "stress-ready" gene expression patterns associated with pre-adaptation to arid environments. Through RNA-seq analysis, we identified orthologous genes between the two species and compared their transcriptomic profiles under both control and drought conditions. Approximately 43% of the orthologous genes exhibited species-specific expression patterns, while nearly 20% were classified as stress-ready. These stress-ready genes were significantly enriched for functions related to nucleosome assembly, RNA metabolism, and transcriptional regulation. Furthermore, transcription factor binding motif analysis revealed a marked enrichment of ERF family motifs, emphasizing their role in both stress-ready and species-specific responses. Our findings indicate that regulatory mechanisms, particularly those mediated by ERF transcription factors, are pivotal to the drought resilience of *S. pennellii*, providing a foundation for future crop improvement strategies.

## Introduction

Plants are sessile organisms that must continuously adapt to survive under environmental stresses [1]. Among these, drought is one of the most significant, affecting essential processes such as germination, photosynthesis, and key metabolic pathways, including carbon and nitrogen assimilation, which ultimately limits agricultural

**Data availability statement:** All relevant data are within the paper and its Supporting Information files. RNA-seq data were retrieved from the Sequence Read Archive database under project number PRJNA800740. Implemented scripts are available on GitHub at https://github.com/JMALab/stress-ready/tree/main.

**Funding:** This work was funded by grants FONDECYT-1210389 and 1250403 (to JMA), FONDECYT-3220673 (to JSCR), FONDECYT-3220801 (to TM), ANID Millennium Nucleus in Data Science and Plant Resilience (PhytoLearning) NCN2024_047, and Instituto Milenio de Biología Integrativa iBio Chile ICN17_002 from the Agencia Nacional de Investigación y Desarrollo de Chile (ANID). JMA's lab is also supported by the National Science Foundation (NSF) Plant Genome Grant NSF-PGRP: IOS-1840761. JMJG is funded by the Ministerio de Ciencia, Innovación y Universidades of Spain (PID2023-151867OB-C33) and the European Union's HORIZON-EIC-PATHFINDEROPEN no. 101098680 (project DARkWIN). This research was partially supported by the supercomputing infrastructure of the NLHPC (ECM-02). The funders had no role in study design, data collection and analysis, decision to publish, or preparation of the manuscript.

**Competing interests:** The authors have declared that no competing interests exist.

production [2]. With climate change exacerbating water scarcity, the threat to crop yields and food security is expected to worsen [3], highlighting the urgent need to develop strategies for crop resilience. Despite ongoing efforts to improve crop tolerance to stress, progress has been slow, largely limited by the complexity of the problem and the scarcity of suitable model systems [4,5]. In this context, plant species that thrive in extreme environments under severe abiotic stress offer valuable insights into the molecular mechanisms of drought adaptation. These species can help identify key regulators and pathways involved in acclimatization to stress [6,7].

One of the most widely cultivated crops globally is tomato (*Solanum lycopersicum*) due to its economic importance [4]. It is a vital agricultural crop especially in arid and semi-arid regions, however, abiotic stresses, particularly those caused by water scarcity, significantly hinder tomato production [8]. At the molecular level, the expression of nearly 4,000 genes changes under drought in tomatoes. These genes are primarily associated with responses to stress, abiotic stimuli, photosystem and oxidative stress, largely mediated by the hormone abscisic acid (ABA) [9–11]. Despite this, some tomato-related species, such as *Solanum pennellii* and *Solanum chilense*, have naturally adapted to water-scarce environments. Native to the semi-arid regions of South America, these species exhibit traits like highly efficient root systems for water uptake, osmotic and physiological adjustments that enable them to endure drought conditions [4,12]. These adaptations make them valuable experimental models for studying key mechanisms that could enhance drought tolerance in commercial tomatoes.

Recent comparative transcriptomics studies of orthologous genes between stress-tolerant and non-tolerant species have revealed distinct gene expression patterns, including shared, unique, and opposing responses [6,7]. Among these patterns, a phenomenon known as the "stress-ready" state has been identified. In this state, the expression level of a gene under non-stress conditions in one species mirrors the expression level of its ortholog under stress conditions in another species [6,7]. Stress-ready genes are constitutively expressed, enabling plants to preemptively prepare for and endure extreme environmental challenges. While previous studies have explored the stress-ready phenomenon in relation to heat and boron stress [6,7], its role in drought adaptation remains largely unexplored in any species.

To unravel it, gene expression patterns in *Solanum lycopersicum* and *Solanum pennellii* were explored by using transcriptomic and protein sequence data to identify responses that may elucidate unique mechanisms of drought adaptation. A systematic approach was applied to identify orthologs between the two species, quantify their expression levels (Fig 1, Step 1), categorize their expression patterns into distinct models, and examine the overrepresentation of biological processes in non-shared responses (Fig 1, Step 2). To uncover key regulatory factors, two complementary analyses were conducted: transcription factor binding motif (TFBM) enrichment in the promoters of orthologs involved in non-shared responses, and a gene regulatory network (GRN) analysis (Fig 1, Step 3).

Throught this workflow, orthologs with shared responses in both species were identified, which are involved in well-characterized pathways such as ABA signaling

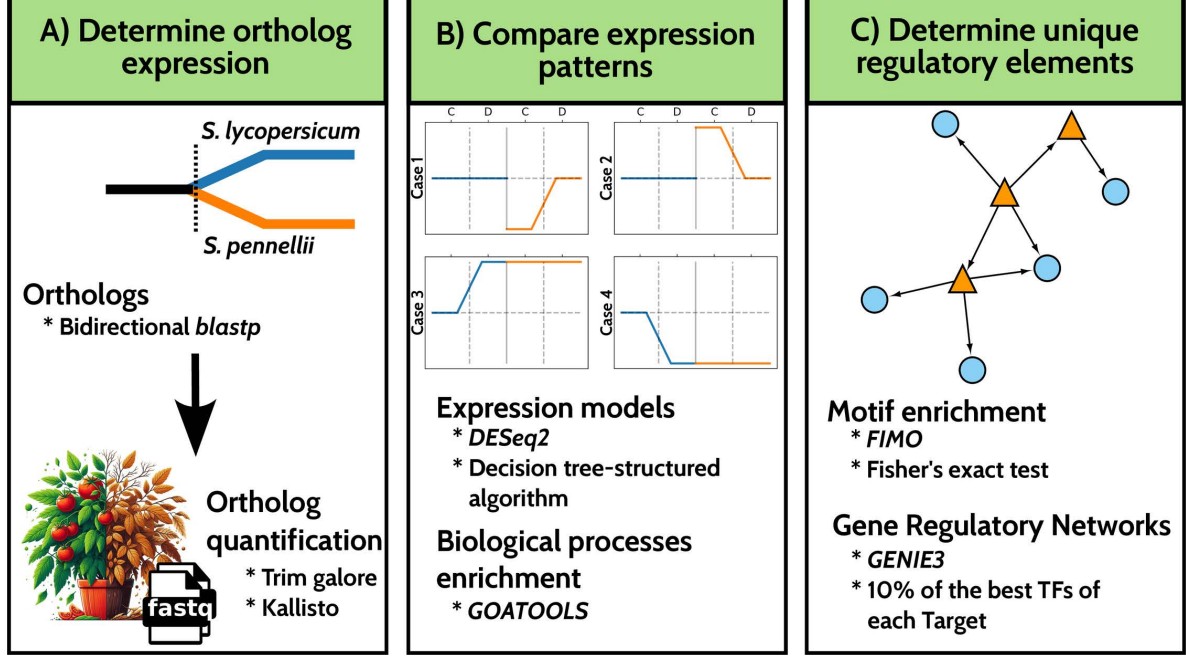

**Fig 1. Workflow for identifying ortholog expression, expression models, and key regulators in *S. lycopersicum* and *S. pennellii*.** (A) Determination of ortholog expression. Orthologs between *S. lycopersicum* and *S. pennellii* were identified using a bidirectional *BlastP* approach. RNA-seq data were processed to estimate the ortholog expression levels in both species under drought conditions using Kallisto. (B) Comparison of expression models. Differential expression analysis was conducted using DESeq2 and custom in-house scripts to identify gene expression patterns across different conditions (control and drought). We define expression models using a decision tree (S1 Fig). Biological processes associated with these expression models were enriched using GOATOOLS for Gene Ontology enrichment. (C) Identification of key regulators. Motif enrichment was performed using FIMO and Fisher's exact test to identify transcription factor binding motifs (TFBMs). Gene regulatory networks (GRNs) were constructed using the GENIE3 algorithm, and the top 10% of transcription factors (TFs) with the strongest regulatory influence on target genes were identified.

and responses to water deprivation. Furthermore, stress-ready states unique to *S. pennellii* were uncovered, including genes involved in nucleosome organization (Case 2) and preactivated genes associated with RNA processing and transcription (Case 3). Additionally, promoters of genes involved in non-shared responses showed significant enrichment for transcription factor (TF) families such as Dof, MIKC/MADS, BBR/BPC, GATA, ERF/AP2, and TCP. Gene regulatory network (GRN) analysis further revealed that the ERF transcription factor subfamily plays a central regulatory role in stress-ready states, emphasizing the importance of differential regulatory mechanisms in plant pre-adaptation to drought.

## Materials and methods

### Orthologs definition and name assignment

To determine orthology among genes of both species, a Basic Local Alignment Search Tool for proteins (BLASTp - V. 2.12.0+) [13] analysis was performed among *S. lycopersicum* and *S. pennellii* proteome V. ITAG3.1 [14] and Spenn v2 [15], respectively using by default parameters and defining orthologs as the best bidirectional *blast* hit [7]. Additionally, to contrast the ortholog assignment, Orthofinder V.2.2.5 [16] was employed with default parameters.

Gene symbol for each tomato identifier were assigned by performing a single BLASTp analysis of *S. lycopersicum* against the *Arabidopsis thaliana* proteome (Araport11 version) [17] deriving each name from the best hit. Then, the orthologs of tomato in *S. pennellii* were assigned the same symbols.

**RNA-seq data analysis**

To analyze the transcriptomic patterns, we used RNA-seq experiments for *S. lycopersicum* and *S. pennellii* under control and drought conditions (11 days of drought treatments). Data was retrieved from the Sequence Read Archive database under project number PRJNA800740 [18]. Once it was downloaded, their sequencing quality was analyzed using *FastQC* V.0.11.9 [19], adapters were removed using *TrimGalore* V.0.6.7 [20], and gene expression quantification was performed using *Kallisto* V.0.46.2 [21] with default parameters and the same genome versions employed in ortholog detection (GenBank accession ID GCA_000188115 for *S. lycopersicum* and GCA_001406875 for *S. pennellii*). Then, to determine quantification by orthologs, we normalized the expression by transcript length in kilobases using equation 1 to test the effects of variations in the size of the ortholog in both species, similarly described by Eshel et al. [6]

$$Q = round\left(\frac{E}{L}\right)$$

(1)

With *Q* represents the discrete quantification value, *E* is the expression level, and *L* is the transcript length in kilobases. Additionally, the expression in its non-length normalized form was also tested in order provide a more comprehensive assessment of potential biases in the quantification process.

**Expression models assignment**

Expression models were fitted based on ortholog expression patterns in *S. lycopersicum* and *S. pennellii* under control and drought conditions. For this purpose, the expression was quantified and normalized by gene length (see RNA-seq Analysis) and differentially expressed ortholog genes were identified using DESeq2 [22], both intra and inter-species, with an adjusted p-value threshold of 0.05.

Then, a rule-based decision tree like algorithm was developed (S1 Fig., S1 File - decision tree and plots.ipynb) using differentially expressed genes and foldchange reported by DESeq2 to categorize ortholog patterns into the following models:

• Stress-ready: Gene expression levels under control conditions in one species match those under drought conditions in the other species (*i.e.*, transcript levels in one plant under drought match those of the other plant under control conditions). There are four cases for this criterion: (Case 1) Gene expression in *S. pennellii* increases under drought to match the levels of *S. lycopersicum* under control conditions; (Case 2) Gene expression in *S. pennellii* decreases under drought to match the levels of *S. lycopersicum* under control conditions; (Case 3) Gene expression in *S. lycopersicum* increases under drought to match the levels of *S. pennellii* under control conditions; (Case 4) Gene expression in *S. lycopersicum* decreases under drought to match the levels of *S. pennellii* under control conditions.

• Shared response: Ortholog expression increases or decreases in both species under drought (Case 1: Upregulated or Case 2: Downregulated).

• Unique response: Expression changes occur only in one species. Upregulated or downregulated in *S. lycopersicum* (Case 1 and 2, respectively) or upregulated or downregulated in *S. pennellii* (Case 3 and 4, respectively).

• Opposite response: Expression changes occur in both species but in opposite directions. Case 1: Gene expression in *S. lycopersicum* is upregulated whilst it is downregulated in *S. pennellii*. Case 2: Gene expression in *S. lycopersicum* is downregulated whilst *S. pennellii* it is upregulated.

• No response: Expression levels remain unchanged under all conditions, with basal levels differing between species (higher or lower in *S. pennellii*, Case 1 and 2, respectively) or being identical (Case 3).

 

### GO term enrichment analysis

The Gene Ontology (GO) terms for tomatoes were obtained from the GOMAP project [23]. These terms were used to conduct an enrichment analysis using the GOATOOLS software V1.4.12 [24] through the *find_enrichment* script using default parameters for each case in expression models, assuming the same term for each ortholog. Finally, we filtered out GO terms that were significantly less concentrated (purified) in the study group compared to the general population and had a False Discovery Rate (FDR) lower than 0.05.

Additionally, in addition to the background comparison, a negative control was performed for each type of response using a random selection of genes, with the same number of genes as those identified for each response, which was carried out in triplicate.

### Motif enrichment in promoters and Gene Regulatory Network modeling

Transcription Factors (TF) binding motifs for *S. lycopersicum* and *S. pennellii* were downloaded from PlantTFDB V.5.0 (https://planttfdb.gao-lab.org/) [25]. TFBM were scanned using the *Find Individual Motif Occurrences* (*FIMO*) V.5.4.1 [26] into promoters of unmasked genomes to retain the full sequence information (2Kb upstream of each gene starting coordinate) with q-value < 0.05. Due to the predictive nature of the information deposited in PlantTFDB, XSTREME V.5.4.1 [27] was employed to identify enriched sequence motifs. This approach enabled us to contrast and validate the findings obtained by FIMO.

To determine motif enrichment in the promoters of genes of interest, a one-tailed Fisher's Exact Test was performed between the group of interest and all other gene groups (e.g., comparing orthologs in stress-ready Case 4 against genes in the shared, unique, opposite, and no response groups). The alternative hypothesis was accepted with a P value < 0.05, and no adjustment method was applied to avoid overlooking potentially important motifs. Additionally, as a negative control, the presence of random motifs was compared against the rest of the genes similarly to the negative control in the GO term enrichment analysis, which was carried out in triplicate.

Finally, gene regulatory networks for both species were developed using the Python version of *GENIE3* [28], with the normalized gene counts from both control and drought conditions as input. A filtering step was then applied, retaining only the top 10% of regulatory factors based on their reported importance. Additionally, edges were kept only if the gene's promoter contained the corresponding TFBM, to enhance network confidence. Once GRNs were obtained, to determine the similarity between them we compute the Jaccard index as described in equation 2 [29].

$$J(E_1, E_2) = \frac{E_1 \cap E_2}{E_1 \cup E_2}$$

(2)

With $E_1$ and $E_2$ representing the sets of directed edges in both networks, and $J$ denote the Jaccard Index, which quantifies the similarity between these GRNs (i.e., edges with the same node pairs and directions) relative to the total number of unique directed edges present in both networks.

## Results

### Orthologous gene expression patterns in response to drought in *S. lycopersicum* and *S. pennellii*

To uncover differential gene expression patterns between the two species under control and drought conditions, it was first necessary to identify orthologous genes. Using bidirectional BLASTp analysis (Fig 1A, S2 Fig.) as previously described [6], we identified 21104 orthologous genes between *S. lycopersicum* and *S. pennellii*. Additionally, we identified 13325 and 23861 unique genes in *S. lycopersicum* and *S. pennellii*, respectively (S2A Fig. and S Table 1). The presence of these non-orthologous genes may be attributed to the high number of proteins with similar sequences within the genome, such as potential paralogs (S3 Fig. A) [30]. Furthermore, Orthofinder comparison identified fewer orthogroups by partitioning

**Table 1. Top ten genes with higher variation under Stress ready (case 2) in *S. pennellii*.**

| Gene *S. pennellii* | Log2(Fold change) | Gene name |
|---|---|---|
| Sopen01g040750 | -1.71493241293427 | GRF zinc finger |
| Sopen09g007800 | -0.566265474557041 | Peptidase family M3 |
| Sopen09g005140 | -0.54605410571863 | Core histone H2A/H2B/H3/H4 |
| Sopen09g031570 | -0.53714497213417 | Core histone H2A/H2B/H3/H4 |
| Sopen03g019890 | -0.515182968238592 | Core histone H2A/H2B/H3/H4 |
| Sopen12g001250 | -0.501431471855018 | Core histone H2A/H2B/H3/H4 |
| Sopen12g006810 | -0.49972363731346 | Core histone H2A/H2B/H3/H4 |
| Sopen01g041030 | -0.492956412903313 | hypothetical protein |
| Sopen12g003870 | -0.462021301453659 | Nucleosome assembly protein (NAP) |
| Sopen05g033890 | -0.409342433125284 | Core histone H2A/H2B/H3/H4 |

these putative paralogs into distinct groups that reflect one-to-one, one-to-many, or many-to-many relationships (S. Fig 3B), thus generating a complex challenge to determine differential expression and assigning gene expression models.

After identifying putative orthologs between the two *Solanum* species, their expression levels were quantified using RNA-seq from Moreira et al. (17) under two conditions: control plants maintained under well-watered conditions and plants subjected to an 11-day drought treatment, in which watering was withheld until the soil reached 20% of field capacity. The Principal Component Analysis (PCA) revealed distinct expression patterns in orthologs, which were influenced not only by species differences, consistent with clade distances [31], but also by environmental conditions, such as drought and control treatments, and corroborated by measuring the Spearman correlation coefficients among replicates (S2B and S2C Fig.)

Furthermore, differential expression analysis showed that a larger number of orthologs are differentially expressed in *S. lycopersicum* compared to *S. pennellii* (6,442 versus 4,049, respectively) (Fig 2A). However, the proportion of upregulated and downregulated genes is consistent within each species, with more genes being downregulated overall and comparable levels of change observed between the two species (Fig 2B and 2C; S2 Table). Additionally, the analysis using data not normalized by gene length yielded similar results, although there were slight variations in p-values and log2foldchange (S3 Table). However, a one-to-one comparison of orthologous genes between the two species is required to distinguish shared and unique responses to drought.

## Stress-ready gene expression highlights *S. pennellii* preadaptation to drought

To identify shared and unique patterns of ortholog expression, differential expression results were analyzed using a rule base algorithm to categorize each ortholog to an exclusive expression model (Fig 1B, S1 Fig., S1 and S2 Tables). Our findings indicate that the majority of orthologs show no significant changes in expression levels under drought, classifying them into the "No response" group. However, among the orthologs with differential expression, most are categorized as either unique or shared responses, comprising approximately 43% and 35% of the total, respectively (Fig 3).

A distinct subset of orthologs exhibits transcriptional preactivation or repression, termed stress-ready. These orthologs are defined as genes whose expression levels in one species under control conditions align with those in another species under stress conditions [6,7]. This group accounts for nearly 20% of all orthologous genes that alter their expression in response to drought, suggesting a form of molecular preadaptation. Fig 3 illustrates the four distinct cases of stress-ready gene expression patterns in orthologous genes between *S. lycopersicum* and *S. pennellii* under control (C) and drought (D) conditions. Case 1: Orthologs in *S. pennellii* under drought conditions reach the expression levels observed in *S. lycopersicum* under control conditions (217 genes, 2.8%). Case 2: Orthologs in *S. pennellii* maintain higher expression levels under control conditions, aligning with *S. lycopersicum* expression under drought conditions (230 genes, 3.0%). Case 3:

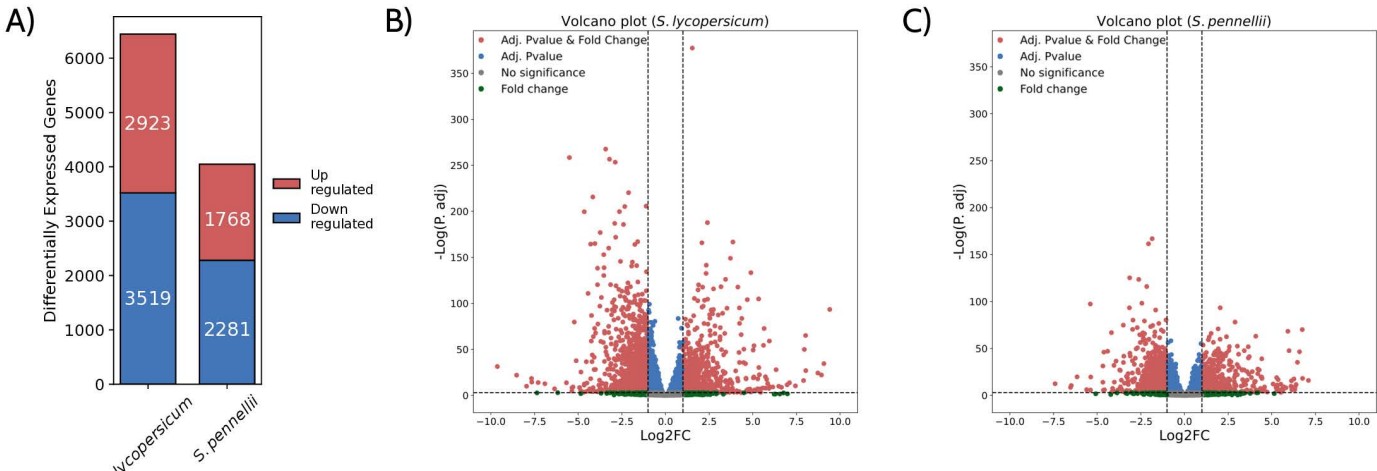

**Fig 2. Comparative analysis of differential gene expression in response to drought between _S. lycopersicum_ and _S. pennellii_.** (A) Bar chart showing the number of differentially expressed genes (DEGs) in response to drought in _S. lycopersicum_ and _S. pennellii_. Upregulated genes are shown in red, and downregulated genes are shown in blue. _S. lycopersicum_ exhibits a higher number of DEGs (6,442 total) compared to _S. pennellii_ (4,049 total), with both species showing more upregulated genes than downregulated ones. (B) Volcano plots displaying the significance (−log10 adjusted p-value) versus fold change (log2 fold change) for DEGs in _S. lycopersicum_ (middle) and _S. pennellii_ (right). Red dots represent genes that meet both fold-change and statistical significance criteria, while blue dots indicate genes significant only by adjusted p-value. Green dots indicate genes with significant fold change but not statistically significant. Grey dots represent genes with no significant changes.

Orthologs in _S. lycopersicum_ under drought conditions reach the expression levels observed in _S. pennellii_ under control conditions (570 genes, 7.4%). Case 4: Orthologs in _S. lycopersicum_ under control conditions align with expression levels observed in _S. pennellii_ under drought conditions (537 genes, 7.0%).

Overall, these results highlight the distinct gene expression responses between the two species, with stress-ready genes constitute nearly one-third of the unique responses under water deprivation, underscoring their importance in drought adaptation. Notably, _S. pennellii_ appears to be preadapted to drought, as evidenced by the higher number of Case 3 and Case 4 orthologs, where _S. lycopersicum_ orthologs align with the expression levels of _S. pennellii_. This suggests that _S. pennellii_ has evolved molecular and regulatory mechanisms enabling it to maintain a stress-ready state, providing a significant advantage in surviving arid environments.

### Enriched biological processes highlight distinct drought-response mechanisms in _S. lycopersicum_ and _S. pennellii_

To gain insights into the biological functions underlying shared and exclusive gene expression patterns, including stress-ready genes, a Gene Ontology (GO) enrichment analysis was conducted (Fig 1B). Within the shared response group, orthologs that exhibited increased expression under drought in both species were enriched for genes associated with seed dormancy, water deprivation, abscisic acid (ABA) response, and oxidative stress, among other processes (S4 Table and S4 Fig). Conversely, orthologs with decreased expression under drought were associated with RNA metabolism, peptide biosynthesis, energy-related processes such as ATP and NADPH biosynthesis, photosynthesis (e.g., Photosystem II), and DNA replication, including regulation of the G2/M transition of the mitotic cell cycle (S4 Table and S4 Fig). These findings align with previous studies in _S. lycopersicum_, which identified similar GO categories as being affected by drought [10,11,32].

In contrast, unique responses showed distinct enrichments depending on the case. For Case 1 (genes induced by drought only in _S. lycopersicum_), enriched functions included mRNA processing, as seen in shared responses, along with histone and RNA modifications (S4 Table and S4 Fig). For Case 2 (genes repressed by drought only in _S. lycopersicum_), was

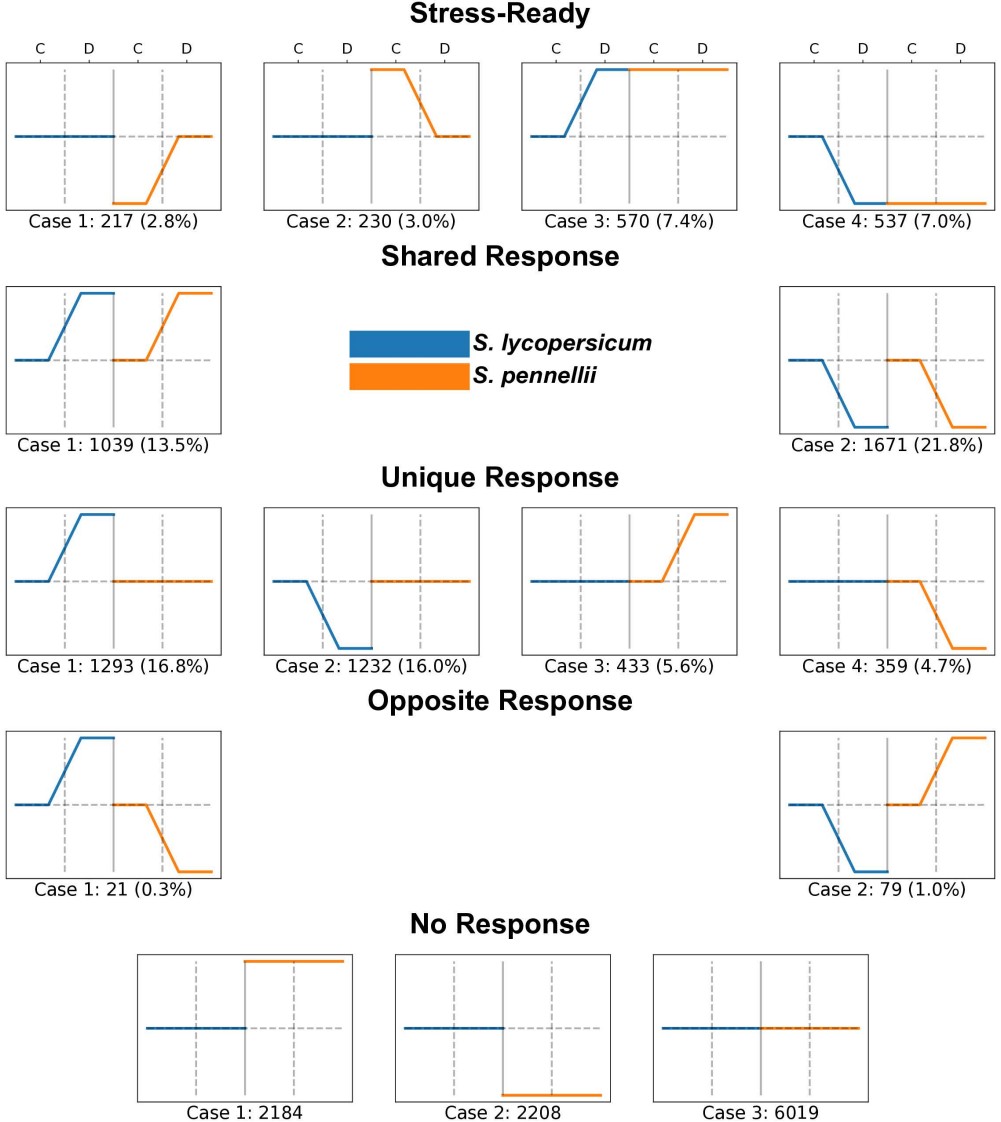

**Fig 3. Distinct drought response patterns in *S. pennellii* highlight unique stress-ready and species-specific mechanisms.** Each line plot shows normalized expression levels for the two species under control (C) and drought (D) conditions. The total number of genes and percentage of the transcriptome for each case is indicated in parentheses. Stress-ready: Genes that are primed for drought in one or both species across four cases, with more pronounced responses in *S. pennellii*. Shared response: Genes exhibiting similar expression patterns under drought in both species across two cases. Unique response: Genes displaying species-specific expression patterns across four cases, particularly highlighting the unique responses of *S. pennellii*. Opposite response: Genes that show inverse expression changes between the two species under drought across two cases. No response: Genes that show minimal or no expression change under drought in each specie across three cases.

primarily related to energy processes, including Photosystem II assembly and repair, NADPH regeneration, and amino acid biosynthesis (S4 Table and S4 Fig). Genes in Case 4 (induced by drought only in *S. pennellii*) were enriched for translation and protein synthesis processes (S4 Table and S4 Fig). Notably, no GO enrichment was found for Case 3 (genes induced by drought only in *S. lycopersicum*). These results highlight distinct drought-response strategies, were *S. lycopersicum* predominantly repress orthologs involved in energy-related processes, while *S. pennellii* uniquely upregulates translation and protein synthesis pathways, suggesting an adaptive mechanism that bolsters protein synthesis under drought stress.

For orthologs classified as stress-ready, significant GO enrichment was observed in Case 2 and 3. In Case 2, where *S. pennellii* orthologs are repressed under drought and match the expression levels of *S. lycopersicum* under control conditions, enriched functions included nucleosome assembly and DNA conformation changes, particularly genes encoding nucleosome histone core components (Table 1, Fig 4A, and S4 Table). For Case 3, where *S. pennellii* orthologs are preactivated and match the drought-induced expression of *S. lycopersicum*, showed enrichment in RNA processing, protein import into the nucleus, and transcription-related processes. Additionally, to verify whether the normalization process and the binary rules applied to the decision tree-like algorithm (e.g., differentially expressed vs. non-differentially expressed genes, positive vs. negative fold-change) effectively distinguish between different gene expression models, their expression patterns were examined (Fig 4B). Within this category, drought response genes that display these specific stress-ready expression patterns were identified (Fig 4C).

Finally, none of those processes were found to be enriched in the negative control (S5 Table), and thus, these findings suggest that stress-ready genes in *S. pennellii* are closely linked to gene expression regulation, playing a key role to plant preadaptation to arid environments. Additionally, the presence of drought-related genes within this group highlights a complex interaction between epigenetic regulation, gene expression, and mechanisms underlying adaptation to adverse environmental conditions (Table 2, Fig 4, and S4 Table).

### *S. pennellii* exhibits differential regulatory elements enriched in unique expression patterns

Since *S. pennellii* exhibits distinct gene expression patterns, including unique and stress-ready states that may contribute to its drought tolerance, transcription factors (TFs) potentially regulating these adaptations were examined. To do so, the FIMO tool was used to determine transcription factor binding motifs (TFBMs) within the promoters of genes grouped in each expression model in both species. A Fisher's exact test was then applied to evaluate the statistical significance of motif enrichment in these responses compared to a genomic background (Fig 1C).

Our analysis revealed that specific TF families were enriched in distinct gene categories. For instance, TFBMs from the GATA family were enriched in the promoters of *S. pennellii* genes categorized under unique response 1. Specifically, three GATA TF motifs were significantly enriched in *S. pennellii* for this expression pattern, while no enrichment was observed in *S. lycopersicum* orthologs (Fig 5A and S5 Fig). GATA motifs were found in the promoters of 65 out of 1,293 orthologs in *S. pennellii* (Fig 5A and 5C). In contrast, two bZIP family motifs were exclusively enriched in *S. lycopersicum* under the unique response 3 category, with an additional bZIP member enriched in the unique response 4 category (Fig 5A and S5 Fig).

For stress-ready orthologs, motifs from the TALE, DOF, and MIKS MADS TF families were exclusively enriched in *S. pennellii* under stress-ready case 3. Interestingly, Trihelix and MYB motifs were enriched in *S. pennellii* promoters under stress-ready case 4. Notably, the ERF family exhibited the highest number of enriched TF motifs, particularly in *S. pennellii* promoters under stress-ready case 3. ERF motifs were present in approximately 12% of the 570 ortholog promoters in *S. lycopersicum*, compared to 20% in *S. pennellii* (Fig 5B and Fig 5C), with De novo motif discovery also recovers ERF family motifs in S. pennellii (S6 Table). Finally, to validate the specificity of TFBM enrichment, Fisher's exact test on triplicate sets of randomly selected promoters was performed. These analyses did reveal some inconsistent motif enrichments, appearing significantly only in one of the three replicates in each model of expression (S7 Table). Although these findings arise from in-silico methods, they underscore the consistency of the predictive data deposited in PlantTFDB and provide additional insight into the observed motif enrichments.

These results suggest that TFs from the ERF, GATA, MYB, and TCP families play pivotal roles in regulating drought-response gene expression. Promoters of *S. pennellii* genes exhibited significant enrichment across multiple unique and stress-ready states, with the ERF family in particular likely contributing to the enhanced stress-adaptive capacity of *S. pennellii* under drought conditions, with TFBM enrichments are likely not merely artifacts of random genomic sampling but potentially reflect genuine regulatory signatures.

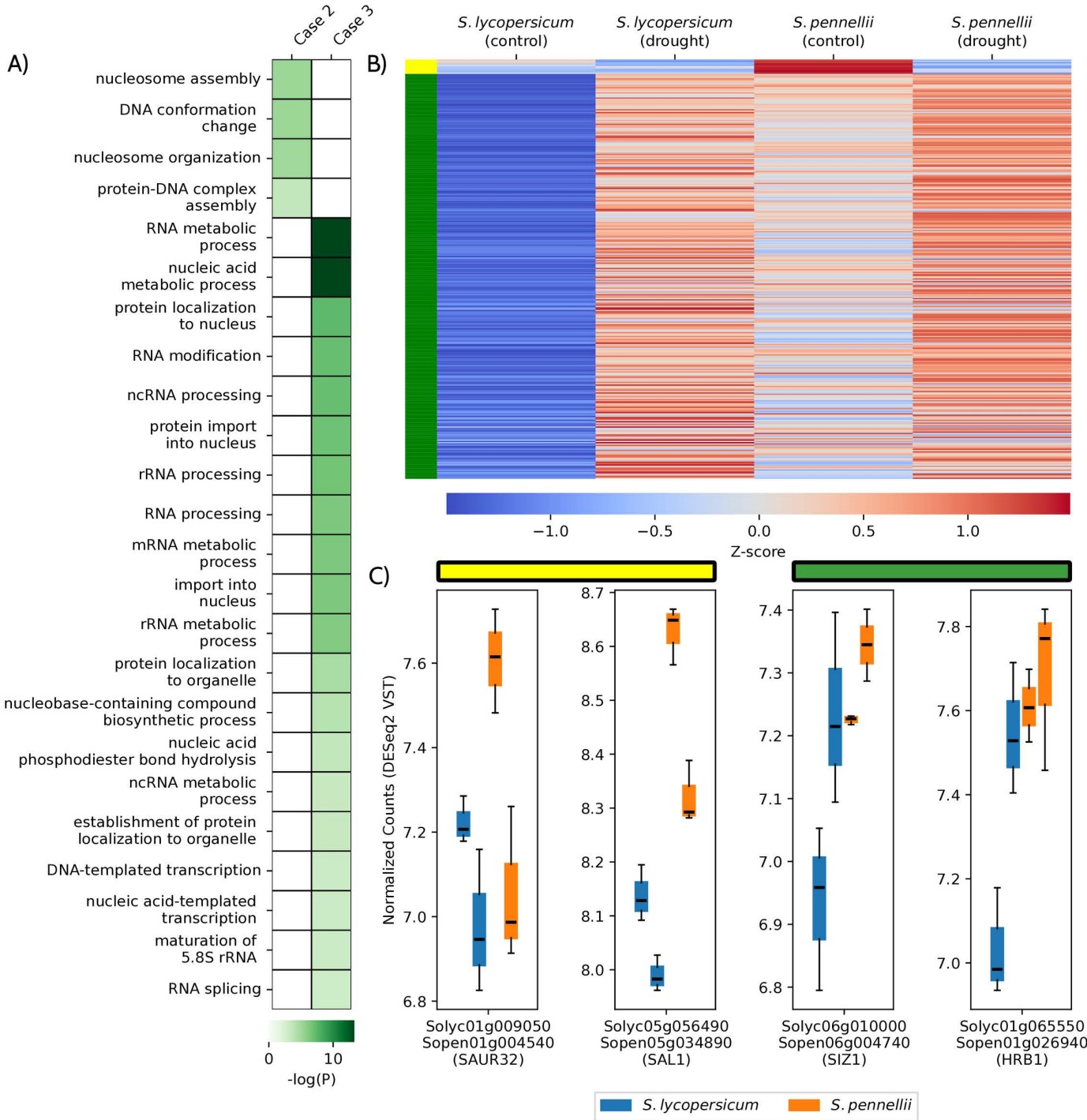

**Fig 4. Stress-ready orthologs exhibit distinct expression patterns, contributing to specific biological processes.** (A) GO term enrichment analysis for stress-ready orthologs in cases 2 and 3, shown on a green scale representing −log10(P value). (B) Mean standardized expression values (Z-scores) for the most enriched processes in the second (yellow) and third (green) cases of stress-ready responses. (C) VST transfiorrmed expression levels of two orthologs from Stress-ready case 2 (yellow bar) and two from case 3 (green bar). These orthologs were selected due to their role under drought reported in Arabidopsis thaliana orthologs.

**Table 2. Top ten genes with higher variation under Stress ready (case 3) in *S. lycopersicum*.**

| Gene *S. lycopersicum* | Log2(Fold change) | Gene Name |
|---|---|---|
| Solyc06g068170 | 5.27650708330517 | U11/U12 small nuclear ribonucleoprotein 25 kDa protein |
| Solyc09g097930 | 5.10540002548879 | Transcription termination factor |
| Solyc09g065490 | 4.88819044802989 | Unknown protein |
| Solyc06g035430 | 2.93134029181532 | T-hook motif nuclear-localized protein 3 |
| Solyc10g018640 | 2.84814961696001 | zinc ion-binding protein |
| Solyc11g073110 | 2.63846052419895 | Protein UPSTREAM OF FLC |
| Solyc02g077840 | 2.17241308436188 | Ethylene-responsive transcription factor 4 |
| Solyc12g044390 | 2.10083049649958 | Ethylene-responsive transcription factor |
| Solyc12g014590 | 1.94167056123487 | Pirin-like protein |
| Solyc06g066020 | 1.7820673459409 | auxin-regulated |

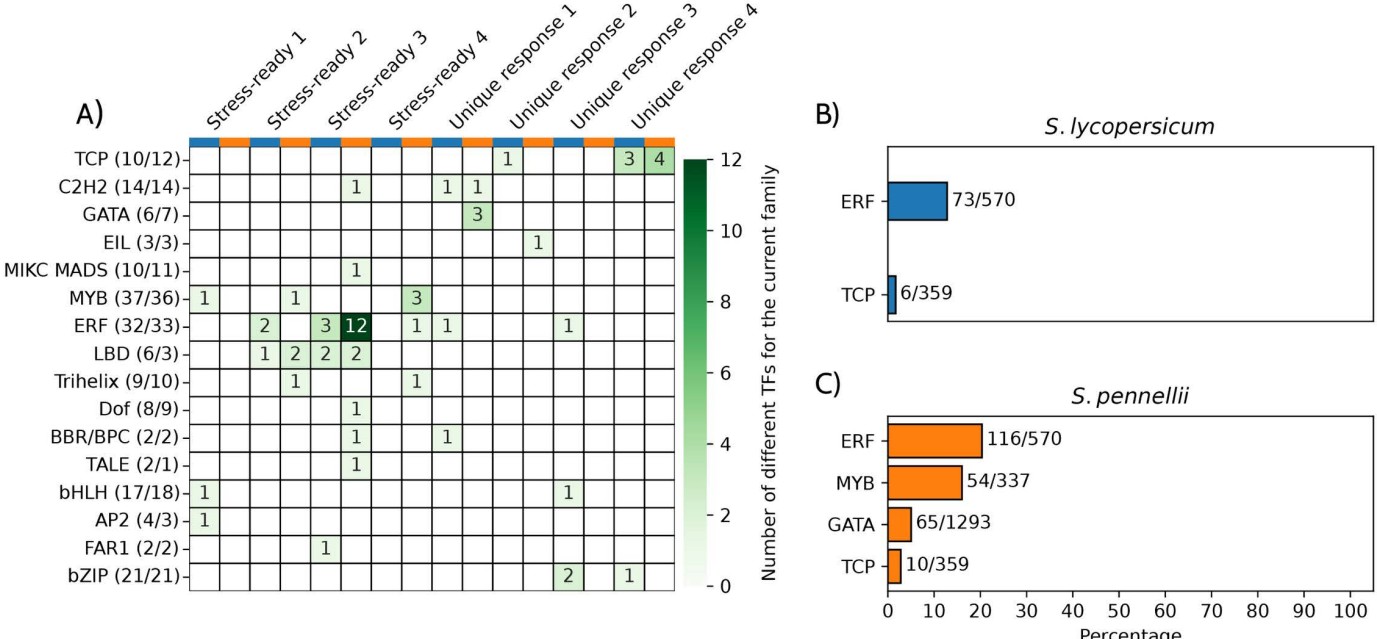

**Fig 5. Differential motif enrichment between gene categories and tomato species.** (A) Heatmap showing the enrichment of transcription factor (TF) families in the promoters of genes categorized under stress-ready and unique response models in *S. lycopersicum* and *S. pennellii*. The x-axis represents different gene expression categories, and the y-axis represents TF families. Numbers in parentheses show the quantity of TFBM family members for each specie (*S. lycopersicum* and *S. pennellii* respectively) deposited in PlantTFDB. Numbers in cells indicate the number of distinct TFs within each family that are enriched in each gene category. The color gradient represents the number of different TFs, with darker shades of green indicating higher numbers of enriched TFs. (B) Bar chart showing the percentage of ERF and TCP TF motifs enriched in *S. lycopersicum* ortholog promoters. (C) Bar chart showing the percentage of ERF, MYB, GATA, and TCP TF motifs enriched in *S. pennellii* ortholog promoters. The numbers above bars indicate the number of enriched TF motifs relative to the total number of gene promoters analyzed for each category.

## ERF transcription factors contribute to the regulation of stress-ready genes

As stress-ready case 3 contains the highest number of stress-ready genes and exhibits specific enrichment of TF motif elements in *S. pennellii*, we further investigated the potential role of individual TFs from the ERF family in regulating genes within this category. Certain TF motifs were shared between the two species, including ESE3 and DEAR2 (Fig 6A).

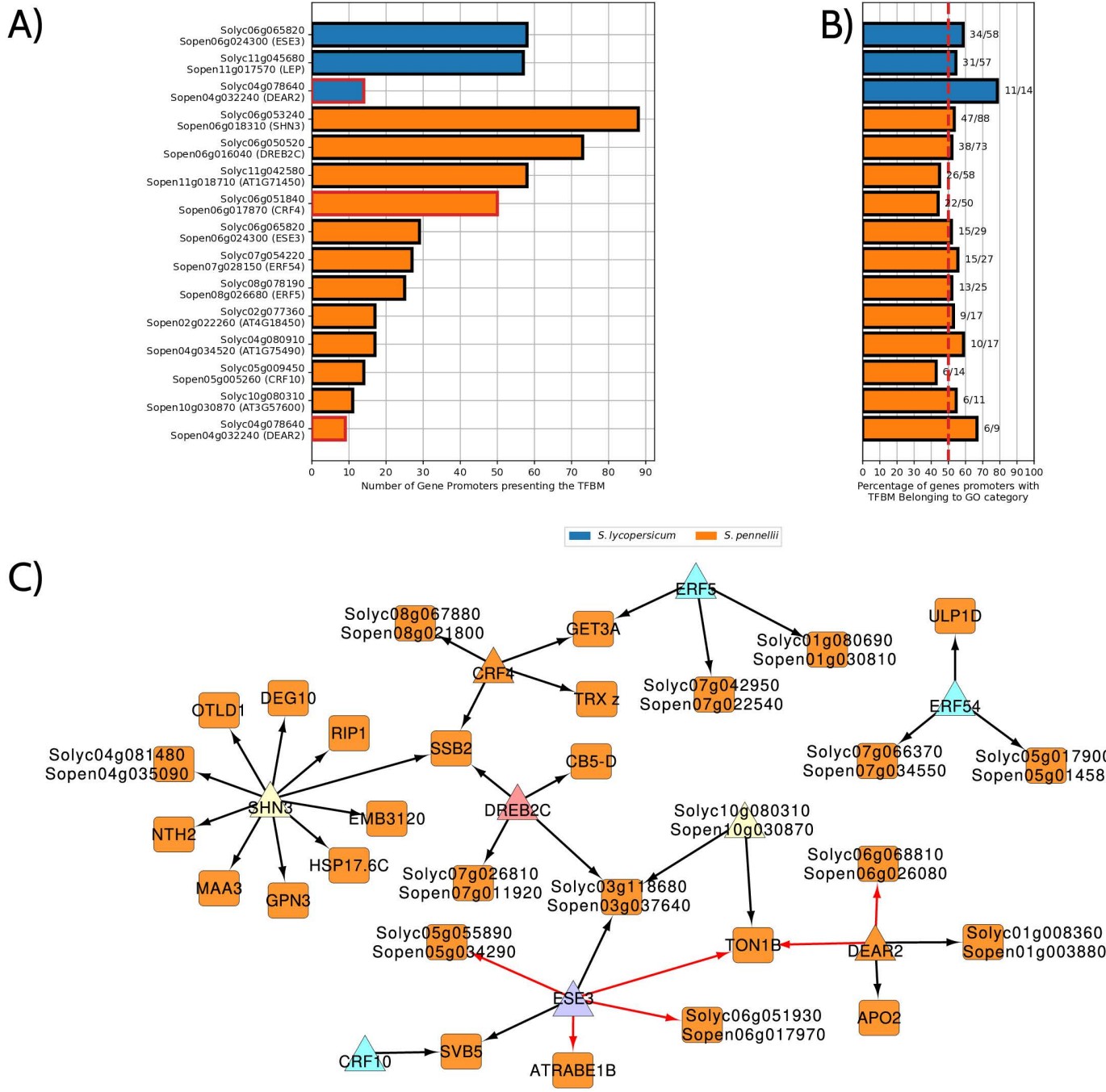

**Fig 6. *S. pennellii* shows greater enrichment of individual ERF TF motifs in gene promoters compared to *S. lycopersicum*.** (A) Bar chart illustrating the number of genes with specific TFBMs in their promoters for both *S. lycopersicum* (blue) and *S. pennellii* (orange). *S. pennellii* has a higher number of genes with motifs associated with drought response, including SHN3, CRF4, and DEAR2, compared to *S. lycopersicum*. (B) Percentage of gene promoters containing TFBMs belonging to specific gene ontology (GO) categories overrepresented in the third category of stress-ready. The red dashed line at 50% highlights that *S. pennellii* consistently exceeds this threshold for drought-related motifs, suggesting a functional association. (C) High-confidence GRN illustrates ERF family members (represented by triangle nodes) regulating stress-ready gene orthologs (represented by rectangle nodes) in *S. pennellii* and *S. lycopersicum* (indicated by black and red edges, respectively). Node colors represent orange for stress-ready case 3, purple for stress-ready case 4, sky blue for no response case 3, yellow for no response case 2, and red for shared response case 1.

The ESE3 motif was found in the promoters of 58 *S. lycopersicum* genes and 29 *S. pennellii* genes, while the DEAR2 motif was detected in 14 *S. lycopersicum* and 9 *S. pennellii* promoters (S6 Fig.), where the higher number of promoters harboring these motifs in tomato could be related to the necessity of increased expression of these genes under drought conditions. Notably, DEAR2 followed the stress-ready Case 3 expression pattern in both species, as highlighted in red in Fig 6A.

Several ERF family TFs were exclusively identified in *S. pennellii*, including SHN3, DREB2C, Sopen11g018710, and CRF4, each targeting 10 or more genes. These TFs were associated with biological functions relevant to stress-ready Case 3, as indicated by GO term enrichment (Figs 4 and 6). Furthermore, nearly 50% or more of the genes containing these TF motifs were involved in RNA metabolic processes and transcriptional responses under drought (Fig 6B), highlighting the critical regulatory role of these TFs in controlling key biological processes.

To explore potential co-regulatory events and determine shared mechanisms between the two *Solanum* species, we constructed high-confidence gene regulatory networks (GRNs) for each species. We identified the top 10% of most relevant TF-target gene pairs using the Random Forest-based algorithm GENIE3 and integrated this with TFBM presence in promoters (Fig 1C). Four ERF members —Solyc11g045680/Sopen11g017570 (LEP), Solyc04g080910/Sopen04g034520, Solyc02g077360/Sopen02g022260, and Solyc11g042580/Sopen11g018710— were excluded due to their low expression levels lower than 10 raw reads counts [22] (S7 Fig). Although the resulting GRNs for *S. pennellii* and *S. lycopersicum* had a similar number of nodes and edges and contained many orthologous TFs (Table 3 and Cytoscape session available on GitHub at https://github.com/JMALab/stress-ready/blob/main/supp.%20file%201%20-%20network.cys), the number of shared regulatory edges was minimal. The Jaccard similarity coefficient was computed to be 0.044 prior to FIMO and 0.019 after its application, indicating that only 1.9% of the directed edges were shared between the two networks. Therefore, these results suggest that, in addition to the importance of each regulation of orthologous genes in each species, there is also a difference in the TFBMs present between the species.

Examining the expression of ERF family members enriched in one or both GRNs revealed that most fall into the "No response" category, suggesting that mechanisms beyond simple up- or downregulation of gene expression may drive their regulatory roles (Fig 6C). Moreover, distinct regulatory events were also observed between the two species. While most target genes were regulated by a single TF, co-regulation was evident for genes with two or more regulators. For example, DREB2C and Sopen10g030870 co-regulate Sopen03g037640, while SHN3 appears to function as a primary independent regulator (Fig 6C). These findings suggest that *S. pennellii* has evolved specialized regulatory mechanisms to control the stress-readiness of orthologous genes, enhancing its ability to tolerate drought conditions.

## Discussion

Drought represents one of the most significant challenges to agriculture, severely affecting crop yields and accelerating the degradation of fertile land areas [3,5]. In response to drought, plants activate diverse molecular mechanisms, with one

**Table 3. Overview of resulting GRNs.**

| | Nodes (acting as TF) | Edges |
|---|---|---|
| *S. lycopersicum* (before FIMO filter) | 17895 (1104) | 1743422 |
| *S. pennellii* (before FIMO filter) | 17780 (1126) | 1759592 |
| Intersection (before FIMO filter) | 17583 (1059) | 150188 |
| Union (before FIMO filter) | 18092 (1171) | 3352826 |
| *S. lycopersicum* (after FIMO filter) | 4955 (60) | 6322 |
| *S. pennellii* (after FIMO filter) | 5059 (55) | 6465 |
| Intersection (after FIMO filter) | 1776 (24) | 236 |
| Union (after FIMO filter) | 8238 (73) | 12551 |

of the most well-characterized being the regulation of stomatal closure through abscisic acid (ABA)-dependent pathway [9]. However, some plants employ unique strategies that allow them to tolerate stress by maintaining a preconditioned state, often referred to as a stress-ready state [7].

Here, a bidirectional BLASTp approach was employed to identify ortholog genes across species followed by gene expression model assignment and finally determining key regulators of stress-ready responses. However, some information could be lost due to failing in ortholog detection in cases such as possible paralogues. Alternative methodologies exist for this purpose such as OrthoFinder [16], a tool that assigns genes to orthogroups—sets of orthologous and paralogous genes descended from a single ancestral gene. Additionally, it build rooted phylogenetic trees for each orthogroup, thereby facilitating comparative genomic analyses. Nevertheless, a limitation of OrthoFinder is that multiple genes per genome are often assigned to the same orthogroup, which generate an interpretative challenge for the differential expression analyses and thus determine their ortholog expression model as shown in S3B Fig., with only 12953 orthogroups contained a single gene per genome, resulting in a data loss exceeding 50% and consequently, bidirectional BLASTp was used for the analyses.

Not all stress-resistant plants exhibit a stress-ready transcriptome [6], and the molecular underpinnings of this state remain poorly understood. Furthermore, it is unclear whether stress-adapted plants universally rely on similar pathways to survive under adverse environmental conditions. In this study, evidence that *S. pennellii* employs unique regulatory mechanisms and a stress-ready transcriptomic state was found, which likely contribute to its remarkable adaptation to arid environments. These findings provide new insights into the diversity of plant responses to drought and highlight the importance of species-specific strategies in stress tolerance. Additionally, we considered including the drought-tolerant species *Solanum chilense*. However, the only publicly available RNA-seq dataset for this species under drought stress (PRJDB15063) uses a different experimental protocol: irrigation was withheld for seven days, then partially resumed, with samples collected on day 12. Due to these methodological differences, particularly in drought duration, recovery phase, and sampling time, direct comparisons under equivalent conditions were not feasible and thus not included in this analysis.

When categorizing genes by expression patterns, we identified well-established GO categories upregulated in both species under drought (Shared response Case 1), including processes related to water deprivation and ABA-mediated signaling, which are critical for plant adaptation to water stress. Similarly, consistent with previous tomato studies, genes downregulated during drought were primarily associated with energy production, reflecting the suppression of photosynthesis under water deficit conditions [4,10]. In contrast, in the unique responses where only *S. lycopersicum* orthologs showed decreased expression, energy-related processes were notably affected (S4 Fig). This suggests an additional vulnerability in *S. lycopersicum* under drought, while maintaining these processes in *S. pennellii* may support higher photosynthetic efficiency, as previously reported [4].

Notably, nearly one-third of drought-responsive orthologs, representing 20% of all orthologs analyzed, were associated with a stress-ready state, suggesting that *S. pennellii* possesses a stress-ready transcriptome. Previous studies on boron stress revealed that 50% and 18% of orthologs in shoots and roots, respectively, exhibited pre-activation or pre-repression states, supporting pre-adaptation mechanisms [7]. In contrast, heat stress studies reported only 6.6% of orthologs in a stress-ready state, with most orthologs displaying shared responses, providing weaker evidence for pre-adaptation in that context [6]. Altogether, while the conservation of these Shared responses points to a conserved stress-responsive modules in both plants, a key distinction lies in the activation dynamics of Stress-ready genes. *S. lycopersicum* only induce these pathways upon stress exposure, whereas the desert-adapted *S. pennellii* maintains these regulatory networks in an anticipatory state even under non-stress conditions, with differences likely manifesting at the regulatory level as illustrated in Fig 6. Future studies should focus on dissecting cis-regulatory elements (promoters, enhancers) to mechanistically explain this evolutionary divergence and potentially harness these insights to enhance crop resilience.

Among the stress-ready groups, only Cases 2 and 3 showed significant enrichment for biological processes. In Case 2, where *S. lycopersicum* orthologs remain unaffected by drought while *S. pennellii* orthologs are repressed, enriched functions were related to nucleosome assembly (Table 1; Fig 4). This suggests a link to epigenetic regulatory mechanisms, particularly histone turnover, which remains less studied compared to post-translational histone modifications like acetylation and methylation [33–37]. Repressing nucleosome assembly genes in *S. pennellii* may stabilize existing epigenetic marks, conserving resources and supporting efficient regulation of drought-responsive gene expression [38]. This aligns with functions related to histone acetylation and methylation observed in *S. lycopersicum* under Unique response Case 1 (S4 Table), suggesting that *S. pennellii* adopts a more energy-efficient strategy during stress [39].

Stress-ready Case 3 orthologs (pre-activated in *S. pennellii*) were enriched for nucleic acid and RNA metabolic processes (S4 Table), crucial for regulating gene expression and transcript adaptation under abiotic stress [40–43]. DNA-template transcription is also enriched in stress-ready Case 3 genes. These findings suggest that *S. lycopersicum* plants activate transcriptional regulation and RNA metabolic genes to meet the demands of drought, while *S. pennellii* employs pre-adaptive mechanisms to bypass these limitations. Similar patterns have been observed in *Schrenkiella parvula*, another species with a stress-ready transcriptome, which may represent a generalized adaptive strategy [7].

In the regulation of stress-ready responses, the ERF TF family plays a critical role in modulating Case 3 stress readiness. ERF proteins are multifunctional regulators known to enhance resilience to various stressors, including drought, salinity, extreme temperatures, and heavy metals. In tomato, the ERF subfamily comprises 77 genes classified into nine subclasses, reflecting considerable evolutionary diversification [44] and depending on which family they proceed, they act in processes such as root hair development, leaf senescence, and proline synthesis for osmoprotection [45]. Given their multifaceted roles, further exploration of genome-wide ERF-mediated regulation in desert-adapted tomatoes is essential.

Interestingly, despite the established importance of ERFs in stress responses, most ERFs associated with Case 3 stress-ready genes displayed "no response" behavior (Fig 6C). This suggests that basal expression levels, rather than transcriptional changes, may be sufficient to drive stress responses, potentially relying on post-translational modifications such as phosphorylation under drought conditions or ubiquitination under normal conditions [46]. Similar mechanisms have been reported for ABF TFs, members of the bZIP family, which rely on phosphorylation sites critical for drought responses without significant changes in transcript levels [47,48]. Additionally, it has been observed that some ERFs can regulate different cis-acting elements such as GCC and DRE boxes, enabling flexible adaptive responses to various abiotic stresses. Particularly notable is the evolutionary conservation in drought-stress response observed in subclass D ERFs between monocotyledonous and dicotyledonous plants, suggesting these factors might represent conserved regulatory elements across plant evolution [49].

Two ERF TFs, CRF4 and DEAR2, were identified as key drivers of Case 3 stress-ready responses. CRF4, enriched exclusively in *S. pennellii*, contains a motif previously linked to cold responses, though its role in drought remains unexplored [50]. DEAR2, active in both species, regulates distinct gene sets, reflecting divergent promoter activity. This mirrors observations in heat-tolerant plants, where rewiring of promoter regions drives species-specific stress tolerance phenotypes [51].

These findings hold significant implications for agricultural biotechnology. Understanding the role of stress-ready orthologs in pre-adaptive responses could enable the development of crops with enhanced stress resilience. By targeting key transcription factors like ERFs and elucidating their role in stress adaptation, it may be possible to engineer crops with enhanced drought resilience, thereby improving agricultural productivity under climate stress.

## Conclusions

Drought is a global challenge with widespread impacts across multiple sectors, underscoring the vulnerability of ecosystems and agriculture to climate change. While plants possess innate mechanisms to mitigate environmental stress, such as stomatal closure to reduce water loss, domesticated crop varieties often lack the adaptive resilience observed in their

wild relatives. This disparity is evident when comparing cultivated species to their undomesticated, evolutionary related counterparts, which retain survival strategies under drought conditions.

This study sheds light on how *S. pennellii*, a wild tomato species native to arid regions, beyond to share ABA mediated pathways with commercial tomato, leverages a constitutively stress-ready transcriptomic state to achieve preadaptation to water deprivation. Comparative genomic analyses revealed that approximately 20% of orthologous genes whose responses are altered under this stress display preadapted expression patterns.

Functional enrichment analysis linked to these genes are associated mainly to nucleosome assembly, RNA metabolism, and transcriptional regulation, suggesting an epigenetic and transcriptional priming mechanism that allows to S. *pennellii* to be preadapted to drought, with the ERF (ethylene-responsive factor) transcription factor family emerging as a critical driver of the stress-ready transcriptomic profile. Notably, the divergence in stress-responsive gene expression between wild and domesticated species may also arise from variations in cis-regulatory regions, which modulate transcriptional activity in a context-specific manner. Future comparative studies should investigate regions such as promoters/enhancers to uncover their role in fine-tuning drought adaptation strategies.

By elucidating these adaptive mechanisms, this research advances our understanding of plant resilience to drought. Future studies should prioritize exploring the translational potential of stress-ready genes and cis-regulatory motifs to engineer crops with enhanced drought tolerance. Such efforts could boost agricultural productivity in water-scarce regions, offering a sustainable strategy to address climate-driven food security challenges.

## Supporting information

**S1 Fig. Decision Tree to Assign Ortholog Expression Model.** The diagram shows a series of binary decisions based on intra- and inter-species gene expression criteria and fold change to categorize ortholog genes into specific expression models. (TIF)

**S2 Fig. Comparison of *S. lycopersicum* and *S. pennellii*.** A) Number of orthologs and genes with and without an ortholog. B) Principal Component Analysis of orthologs expression under control and drought conditions, showing a clear separation between specie (Principal Component 1) and condition (Principal Component 2). C) of orthologs expression under control and drought conditions. (TIF)

**S3 Fig. Gene alignment and Orthofinder analysis between *S. lycopersicum* and *S. pennellii*.** A) Graph shows the Number and percentage of genes with and without alignment in the other species (red and blue bar respectively). The results highlight a high proportion of genes in each species with corresponding homologous genes in the other species. B) Classification of orthologous gene relationships given by Orthofinder, depicting the distribution of one-to-one, one-to-many, and many-to-many orthologous gene mappings between *S. lycopersicum* and *S. pennellii*. (TIF)

**S4 Fig. Selection of enriched GO terms for orthologs associated with shared and unique responses.** The heatmap highlights that most biological processes (rows) are distinct, with partial overlap observed in RNA metabolic processes and energy functions between responses. Green shades indicate −log10(adjusted P-value). (TIF)

**S5 Fig. Heatmap of Fisher's exact test p-values for TFBM enrichment across promoters of orthologs in each expression model.** Rows represent TF ortholog in both *S. lycopersicum* and *S. pennellii*, along with their respective TF families, while columns correspond to the different stress-ready and unique response cases in *S. lycopersicum* (blue rectangles) and *S. pennellii* (orange rectangles). Green heatmap indicates p-values on a -log scale. (TIF)

**S6 Fig. Presence of binding motifs in promoters of ortholog genes for the ERF transcription factors shared between species.** The upper panel shows the frequency of occurrence in the promoters, while the lower panel shows the position for each ortholog.
(TIF)

**S7 Fig. Heatmap of expression model groups for each enriched TFBM.** Rows represent TFBMs enriched in promoters, as shown in S5 Fig. Columns correspond to the different stress-ready and unique response cases in *S. lycopersicum* (blue rectangles) or *S. pennellii* (orange rectangles). Green shades indicate the group affiliation of the current TF.
(TIF)

**S1 File. Script to perform Gene Expression Model Assignment.**
(ZIP)

**S1 Table. BLASTp results for ortholog definition and gene symbol assignment.** This table contains the results of the bidirectional BLAST between *S. lycopersicum* and *S. pennellii*, the non-bidirectional BLAST results, and the gene assignment based on BLAST between tomato and *Arabidopsis thaliana.*
(XLSX)

**S2 Table. DESeq2 outputs for gene length normalization.** This table contains multiple intra- and inter-species comparisons, contrasting control and drought conditions.
(XLSX)

**S3 Table. DESeq2 outputs without gene length normalization.** This table contains multiple intra- and inter-species comparisons, contrasting control and drought conditions.
(XLSX)

**S4 Table. GO term enrichment for the Gene Expression Models.**
(XLSX)

**S5 Table. GO term enrichment for random Gene Expression Models assignment.**
(XLSX)

**S6 Table. De novo identification of motifs and their association with binding motifs in PlantTFDB.**
(XLSX)

**S7 Table. Motif enrichment for random Gene Expression Models assignment.**
(XLSX)

## Author contributions

**Conceptualization:** J. Sebastian Contreras-Riquelme, Miguel Contreras, Jose M. Alvarez.

**Data curation:** J. Sebastian Contreras-Riquelme.

**Formal analysis:** J. Sebastian Contreras-Riquelme, Miguel Contreras, Rachid Sjoberg.

**Funding acquisition:** Jose M. Alvarez.

**Investigation:** J. Sebastian Contreras-Riquelme, Miguel Contreras, Rachid Sjoberg.

**Methodology:** J. Sebastian Contreras-Riquelme, Miguel Contreras, Tomas C. Moyano, Jose M. Alvarez.

**Resources:** Jose Jimenez-Gomez.

**Software:** J. Sebastian Contreras-Riquelme.

**Supervision:** Jose Jimenez-Gomez, Jose M. Alvarez.

**Validation:** J. Sebastian Contreras-Riquelme.

**Writing – original draft:** J. Sebastian Contreras-Riquelme.

**Writing – review & editing:** J. Sebastian Contreras-Riquelme, Tomas C. Moyano, Jose Jimenez-Gomez, Jose M. Alvarez.

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
