## [Decision Letter · Decision Letter 0]

28 Feb 2025

PONE-D-25-00360Desert-adapted tomato Solanum pennellii exhibit unique regulatory elements and stress-ready transcriptome patterns to drought.PLOS ONE

Dear Dr. Alvarez,

Thank you for submitting your manuscript to PLOS ONE. After careful consideration, we feel that it has merit but does not fully meet PLOS ONE’s publication criteria as it currently stands. Therefore, we invite you to submit a revised version of the manuscript that addresses the points raised during the review process.

The manuscript was reviewed by two independent experts in the field. Both the reviewers found the work interesting but raised several issues, which should be addressed for further consideration. The reviewers provide detailed comments in their reviews and point out the areas where the manuscript needs to be improved.

We look forward to receiving your revised manuscript.

Kind regards,

Mohammad Irfan, Ph.D.

Academic Editor

PLOS ONE

Journal Requirements:

“This work was funded by grants FONDECYT-1210389 (JMA) FONDECYT-3220673 (to JSCR), FONDECYT-3220801 (to TM), ANID Millennium Nucleus in Data Science and Plant Resilience (PhytoLearning) NCN2024_047, and Instituto Milenio de Biología Integrativa iBio Chile ICN17_002 from “Agencia Nacional de Investigación y Desarrollo de Chile (ANID)”. JMA lab is also supported by National Science Foundation (NSF) Plant Genome Grant NSF-PGRP: IOS-1840761. JMJG is funded by Ministerio de Ciencia Inovación y Universidades of Spain (PID2023-151867OB-C33) and European Union's HORIZON-EIC-PATHFINDEROPEN no. 101098680 (project DARkWIN). This research was partially supported by the supercomputing infrastructure of the NLHPC (ECM-02).”

4. Please include captions for your Supporting Information files at the end of your manuscript, and update any in-text citations to match accordingly. Please see our Supporting Information guidelines for more information: http://journals.plos.org/plosone/s/supporting-information .

5. We are unable to open your Supporting Information file supp. file 1 - network.cys and Supp. file2 - scripts.zip. Please kindly revise as necessary and re-upload.

Reviewers' comments:

Reviewer's Responses to Questions

**Comments to the Author**

1. Is the manuscript technically sound, and do the data support the conclusions?

Reviewer #1: Partly

Reviewer #2: Yes

2. Has the statistical analysis been performed appropriately and rigorously? 

Reviewer #1: Yes

Reviewer #2: I Don't Know

3. Have the authors made all data underlying the findings in their manuscript fully available?

Reviewer #1: Yes

Reviewer #2: Yes

4. Is the manuscript presented in an intelligible fashion and written in standard English?

Reviewer #1: Yes

Reviewer #2: Yes

5. Review Comments to the Author

Reviewer #1: The manuscript by Contreras et al present a bioinformatics comparison of gene regulation in two close Solanum species using public data from a previous paper of their own. I like the style of the paper, straight to the point, but still I have comments for the authors:

L127 Please add FASTQC version

L129 Please add GCA assembly codes for both genomes

L129-131 Please rephraser and consider adding equation, in its current form it is not clear what's being done

L133 Doesn't it make more sense to fit instead of assign?

L135 Why were normalized expression values not computed by DEseq or kallisto, for instance in TPMs, why is the current approach used instead? Please justify. Also, are these the same values in Y-axis of Fig4B and Fig4C?

L138 Please provide source code/or a URL to code for the decision tree and explan also how FC is computed

L165 Please add GOATOOLS version

L173 Please add PLANTTFDB version and URL it possible

L174 Please explain the source of TSS or correct if you mean gene start; also explain whether raw or masked genomic sequences were used to scan motifs

L198 The authors should compare their obtained total orthologues to other estimates in the literature or databases to give the reader a sense of how accurate their simple approach is; in particular, as they reckon, the reciprocal blast way is highly sensitive to paralogues and multiple copies. Same problem would apply to assigning Araport gene ids, which I don't get why it was required (sorry if I missed it).

L211 Please help us understand these numbers: is this caused by the different number of genes in both species? Shouldn't both tests produce similar numbers of DEGs? Can this be explained by different variances across replicates?

L248 "minimal or no change" does not match FC > 0. Please see L138

L274 Did the authors compute enrichment for random gene groups (negative control). These should increase the confidence on this section and would help put in context L294.

L307 Why was it neccessary to verify it, how often the decision groups genes with different responsed? I apologize if I misunderstood this point.

L339 See L274, same idea here; how often random gene lists are enriched in motifs? Moreover, please let us know how many motifs from each family were scanned so that readers can gauge whether they were all equally likely to start with. Perhaps this could go in Fig5. Moreover, as motifs in PLANTFDB are mostly predictions, it is possible that some motifs are matched more that others due to differences in motif quality. This limitation should be discussed. I think this part of the work would greatly improve if the most significant motifs reported were confirmed by an ab initio approach, for instance based on the over-representation of DNA sequences, without prior knowledge of the motifs.

L378 It would help to see shared motifs as cis-elements mapped along orthologous promoters side to side, so readers can evaluate the conservation

L414 I find it hard to understand that the GRN are so different, is it because the number of shared cis-elements across orthologous is minimal? Perhaps this number should go into table 3 somehow. Please explain.

Reviewer #2: MAJOR ISSUES:

1. Why was Solanum chilense not considered for this study?

2. There are many instances where first person tone has been used for writing. Please consider using the third person, this is more formal and acceptable.

3. In many instances, there is no consistency in the use of terminology.

4. In many instances, some sentences are long and complex, which might hinder readability. Consider breaking down complex sentences into simpler ones for better clarity.

5. The discussion of ERF transcription factors and their roles in stress responses is good. However, it might benefit from a more detailed explanation of how these factors mechanistically contribute to stress resilience.

6. There is should a section for conclusion and future directions. Which is missing.

MINOR ISSUES:

1.Line 114: "BLASTp" should be defined for readers unfamiliar with it, perhaps as "Basic Local Alignment Search Tool for proteins (BLASTp)."

2.Line 198-201: The numbers of unique genes are inconsistent. If 21,104 is the number of orthologous genes, the unique genes should differ. Double-check these figures for accuracy. Consider rephrasing for clarity: "Additionally, we identified 13,325 unique genes in S. lycopersicum and 21,104 in S. pennellii."

3.Line 202-203: The explanation for non-orthologous genes is clear but consider adding more detail on how paralogs might affect the analysis.

4. Line 204-205: Specify the conditions of the "control" and "drought" treatments for clarity.

5. Line 206: Moreira et al. (2022) is not properly cited.

6. Line 207-209: The sentence is complex; consider breaking it into two for clarity. For example, "PCA revealed distinct expression patterns in orthologs. Species differences influenced these patterns, consistent with clade distances, and by environmental conditions such as drought."

7. Line 210-214: The comparison of differentially expressed genes is clear but ensure that the methodology for determining differential expression is described elsewhere in the document. Consider adding a brief discussion on the biological significance of the findings, especially regarding the implications of differential gene expression in response to drought.

8. Line 231-233: Consider briefly explaining why the decision-tree algorithm method was chosen and how it categorizes orthologs.

9. Lines 266-273: The statement "S. pennellii appears to be preadapted to drought" is a strong conclusion. Consider discussing potential evolutionary mechanisms or historical environmental conditions that may have contributed to this preadaptation.

10. Lines 279-284: Consider rephrasing for clarity: "Orthologs with increased expression under drought in both species were enriched for genes involved in seed dormancy, water deprivation, response to abscisic acid (ABA), and oxidative stress."

11.Lines 333-339: Consider rephrasing for clarity: "Using the FIMO tool, we scanned transcription factor binding motifs (TFBMs) in the promoters of genes associated with each expression model in both species.

12.Line 432: "pathways pathway" is a repetition. It should be corrected to "pathways."

13. Line 454: The sentence structure is awkward: "Notably, nearly one-third of drought-responsive orthologs were associated with a stress-ready state, representing 20% of all orthologs analyzed." Consider rephrasing for clarity: "Notably, nearly one-third of drought-responsive orthologs, representing 20% of all orthologs analyzed, were associated with a stress-ready state."

14. Line 460: The phrase "providing weaker evidence for pre-adaptation in that context r (6)" contains a typographical error with the letter "r." It should be removed.

15. Line 471: The phrase "but S. pennellii appears to adopt a more energy-efficient strategy during stress" could be more concise: "suggesting that S. pennellii adopts a more energy-efficient strategy during stress."

16. Line 481: The sentence is split awkwardly: "ERF proteins are multifunctional regulators known to enhance resilience to various stressors, including drought, salinity, extreme temperatures, and heavy metals, and are involved in processes such as root hair development, leaf senescence, and proline synthesis for osmoprotection (42)." Consider breaking it into two sentences for clarity.

7. PLOS authors have the option to publish the peer review history of their article (https://journals.plos.org/plosone/s/editorial-and-peer-review-process#loc-peer-review-history). If published, this will include your full peer review and any attached files. Do you want your identity to be public for this peer review? YES

7a. If you answered "Yes" above, that you would like your identity to be public. Vishnutej Ellur

6. PLOS authors have the option to publish the peer review history of their article (what does this mean? ). If published, this will include your full peer review and any attached files.

**Do you want your identity to be public for this peer review?** For information about this choice, including consent withdrawal, please see our Privacy Policy .

Reviewer #1: No

Reviewer #2: **Yes: ** Vishnutej Ellur

---

## [Author Response · Author response to Decision Letter 0]

14 Apr 2025

Mohammad Irfan, Ph.D.

Academic Editor

PLOS ONE

Dear Editor

We sincerely thank the editor for the opportunity to resubmit and the reviewers for their valuable comments and suggestions regarding our manuscript (ID: PONE-D-25-00360, titled “Desert-adapted tomato Solanum pennellii exhibit unique regulatory elements and stress-ready transcriptome patterns to drought”). We have thoroughly addressed each point raised and revised the manuscript accordingly and, in addition, we believe that the revisions have significantly enhanced the clarity and overall quality of the manuscript.

Additionally, as suggested, we would like to modify our financial disclosure as follows

"This work was funded by grants FONDECYT-1210389 and 1250403 (to JMA), FONDECYT-3220673 (to JSCR), FONDECYT-3220801 (to TM), ANID Millennium Nucleus in Data Science and Plant Resilience (PhytoLearning) NCN2024_047, and Instituto Milenio de Biología Integrativa iBio Chile ICN17_002 from the Agencia Nacional de Investigación y Desarrollo de Chile (ANID). JMA's lab is also supported by the National Science Foundation (NSF) Plant Genome Grant NSF-PGRP: IOS-1840761. JMJG is funded by the Ministerio de Ciencia, Innovación y Universidades of Spain (PID2023-151867OB-C33) and the European Union's HORIZON-EIC-PATHFINDEROPEN no. 101098680 (project DARkWIN). This research was partially supported by the supercomputing infrastructure of the NLHPC (ECM-02). The funders had no role in study design, data collection and analysis, decision to publish, or preparation of the manuscript."

Below, we provide detailed responses to the reviewers' comments. Please note that all line and page references in our responses correspond to the revised version of the manuscript without tracked changes.

PONE-D-25-00360

Desert-adapted tomato Solanum pennellii exhibit unique regulatory elements and stress-ready transcriptome patterns to drought.

PLOS ONE

Dear Dr. Alvarez,

Thank you for submitting your manuscript to PLOS ONE. After careful consideration, we feel that it has merit but does not fully meet PLOS ONE’s publication criteria as it currently stands. Therefore, we invite you to submit a revised version of the manuscript that addresses the points raised during the review process.

The manuscript was reviewed by two independent experts in the field. Both the reviewers found the work interesting but raised several issues, which should be addressed for further consideration. The reviewers provide detailed comments in their reviews and point out the areas where the manuscript needs to be improved.

We have included it.

We have included it.

We have included it.

We look forward to receiving your revised manuscript.

Kind regards,

Mohammad Irfan, Ph.D.

Academic Editor

PLOS ONE

Journal Requirements:

“This work was funded by grants FONDECYT-1210389 and 1250403 (JMA) FONDECYT-3220673 (to JSCR), FONDECYT-3220801 (to TM), ANID Millennium Nucleus in Data Science and Plant Resilience (PhytoLearning) NCN2024_047, and Instituto Milenio de Biología Integrativa iBio Chile ICN17_002 from “Agencia Nacional de Investigación y Desarrollo de Chile (ANID)”. JMA lab is also supported by National Science Foundation (NSF) Plant Genome Grant NSF-PGRP: IOS-1840761. JMJG is funded by Ministerio de Ciencia Inovación y Universidades of Spain (PID2023-151867OB-C33) and European Union's HORIZON-EIC-PATHFINDEROPEN no. 101098680 (project DARkWIN). This research was partially supported by the supercomputing infrastructure of the NLHPC (ECM-02).”

We have added the statement “The funders had no role in study design, data collection and analysis, decision to publish, or preparation of the manuscript” in our cover letter.” Thanks for including this in the online submission.

Thanks so much for the clarification. Please modify our funding including the new grant and the following statement: "The funders had no role in study design, data collection and analysis, decision to publish, or preparation of the manuscript." As provided in the previous paragraph

We confirm that the corresponding author’s ORCID iD (0000-0002-5073-7751) has been added and validated in Editorial Manager as instructed.

We have included it according to the guidelines.

5. We are unable to open your Supporting Information file supp. file 1 - network.cys and Supp. file2 - scripts.zip. Please kindly revise as necessary and re-upload.

To make this information available, we have repackaged the files and uploaded them to GitHub (https://github.com/JMALab/stress-ready) along with the code for this manuscript. Please note that the .cys file is provided exclusively in our GitHub repository due to system limitations regarding the number of characters in file names. Additionally, we have replaced the Supp. File 2 scripts with a notebook (S1 File) that can be opened using Jupyter or Google Colab.

Reviewers' comments:

Reviewer's Responses to Questions

Comments to the Author

1. Is the manuscript technically sound, and do the data support the conclusions?

Reviewer #1: Partly

Reviewer #2: Yes

2. Has the statistical analysis been performed appropriately and rigorously?

Reviewer #1: Yes

Reviewer #2: I Don't Know

3. Have the authors made all data underlying the findings in their manuscript fully available?

Reviewer #1: Yes

Reviewer #2: Yes

4. Is the manuscript presented in an intelligible fashion and written in standard English?

Reviewer #1: Yes

Reviewer #2: Yes

5. Review Comments to the Author

Reviewer #1: The manuscript by Contreras et al present a bioinformatics comparison of gene regulation in two close Solanum species using public data from a previous paper of their own. I like the style of the paper, straight to the point, but still I have comments for the authors:

Thanks to the reviewer for the positive comments.

L127 Please add FASTQC version

We have specified it in the manuscript (FastQC V.0.11.9).

L129 Please add GCA assembly codes for both genomes

We have specified it in the manuscript (GenBank accession ID GCA_000188115 for S. lycopersicum and GCA_001406875 for S. pennellii).

L129-131 Please rephraser and consider adding equation, in its current form it is not clear what's being done

We have rephrased the text and added the equation to describe the methodology more explicitly.

L133 Doesn't it make more sense to fit instead of assign?

We have not used the word "fit," but rather "assignment," because it is an algorithm with a rule-based decision tree structure. For this reason, the text was changed to clearly reflect this difference. Thank you very much for noticing that.

L135 Why were normalized expression values not computed by DEseq or kallisto, for instance in TPMs, why is the current approach used instead? Please justify. Also, are these the same values in Y-axis of Fig4B and Fig4C?

Thanks for this comment. Our approach was guided by the study by Eshel et al, in which stress ready states were also assessed (https://doi.org/10.1111/nph.18411). For this reason, we replicated the normalization method they proposed, which involves potential variations in gene lengths. Since this normalization could influence the classification or assignment of expression models, we also performed an alternative analysis in which we did not normalize by gene length, but only used DESeq2 normalization. This resulted in the same gene assignment; however, there were little variations in log2 fold change and adjusted and unadjusted p-values. Therefore, we have included the new DESeq2 outputs as supplementary material, and this has been mentioned in the manuscript (Line 136, page 7 and Line 236 page 10). Additionally, we have described the values for Y axis in Figure 4B and C.

L138 Please provide source code/or a URL to code for the decision tree and explan also how FC is computed

We have clarified in the text that the fold change (FC) values were derived using DESeq2. The tree-based algorithm (provided in Supplementary File 2: decision_tree_and_plots.ipynb) has now been referenced in the main text.

L165 Please add GOATOOLS version

We have specified it in the manuscript (GOATOOLS V1.4.12)

L173 Please add PLANTTFDB version and URL it possible

We have specified it in the manuscript (PlantTFDB V.5.0 - https://planttfdb.gao-lab.org/)

L174 Please explain the source of TSS or correct if you mean gene start; also explain whether raw or masked genomic sequences were used to scan motifs

We thank the reviewer for raising this point. The text has been modified to clarify that we used gene start coordinates. Also, raw genomic sequences were determined for motif detection to preserve original genomic context.

L198 The authors should compare their obtained total orthologues to other estimates in the literature or databases to give the reader a sense of how accurate their simple approach is; in particular, as they reckon, the reciprocal blast way is highly sensitive to paralogues and multiple copies. Same problem would apply to assigning Araport gene ids, which I don't get why it was required (sorry if I missed it).

We acknowledge the importance of comparing our results to external references. To address this, we compared the orthologs obtained using OrthoFinder with those obtained from other methods such as reciprocal BLAST. OrthoFinder yielded a smaller number of orthologs (orthogroups), many of which showed complex many-to-many relationships, likely due to the presence of paralogs. While this approach may underestimate the total number of orthologs compared to databases or other methods, it provides a more conservative and reliable framework for downstream expression analysis, where one-to-one orthologs are preferable to avoid ambiguity. We have clarified this point in the main text (Line 116 page 6, Line 220 page 10 and Line 478 page 20).

Regarding the BLAST comparison with Arabidopsis, this was used exclusively to assign gene symbols to tomato genes for ease of interpretation. This step was not used for ortholog detection, but only for annotation purposes. We will ensure this is clearly stated in the manuscript to avoid confusion (Line 118 page 6).

L211 Please help us understand these numbers: is this caused by the different number of genes in both species? Shouldn't both tests produce similar numbers of DEGs? Can this be explained by different variances across replicates?

Thank you for your thoughtful question. The difference in the number of DEGs between S. lycopersicum and S. pennellii is not due to the total number of genes annotated in each species, which are comparable. Instead, the variation in DEG counts likely reflects intrinsic biological differences in how each species responds transcriptionally to drought stress. This is consistent with the findings reported by Egea et al. 2018 (https://doi.org/10.1038/s41598-018-21187-2), who observed distinct transcriptomic responses under similar conditions.

To rule out the possibility that differences in DEG counts were due to technica

---

## [Decision Letter · Decision Letter 1]

30 Apr 2025

Desert-adapted tomato Solanum pennellii exhibit unique regulatory elements and stress-ready transcriptome patterns to drought.

PONE-D-25-00360R1

Dear Dr. Alvarez,

We’re pleased to inform you that your manuscript has been judged scientifically suitable for publication and will be formally accepted for publication once it meets all outstanding technical requirements.

Kind regards,

Mohammad Irfan, Ph.D.

Academic Editor

PLOS ONE

Additional Editor Comments (optional):

Reviewers' comments:

Reviewer's Responses to Questions

**Comments to the Author**

1. If the authors have adequately addressed your comments raised in a previous round of review and you feel that this manuscript is now acceptable for publication, you may indicate that here to bypass the “Comments to the Author” section, enter your conflict of interest statement in the “Confidential to Editor” section, and submit your "Accept" recommendation.

Reviewer #1: All comments have been addressed

Reviewer #2: All comments have been addressed

2. Is the manuscript technically sound, and do the data support the conclusions?

Reviewer #1: Yes

Reviewer #2: Yes

3. Has the statistical analysis been performed appropriately and rigorously? 

Reviewer #1: Yes

Reviewer #2: I Don't Know

4. Have the authors made all data underlying the findings in their manuscript fully available?

Reviewer #1: Yes

Reviewer #2: Yes

5. Is the manuscript presented in an intelligible fashion and written in standard English?

Reviewer #1: Yes

Reviewer #2: Yes

6. Review Comments to the Author

Reviewer #1: The authors have significantly improved the paper by clarifying the points raised by both referees, congrats

Reviewer #2: I am happy with the changes made. I will recommend accepting the manuscript. My suugestions were considered and answered.

7. PLOS authors have the option to publish the peer review history of their article (what does this mean? ). If published, this will include your full peer review and any attached files.

**Do you want your identity to be public for this peer review?** For information about this choice, including consent withdrawal, please see our Privacy Policy .

Reviewer #1: **Yes: ** Bruno Contreras Moreira

Reviewer #2: No

---

## [Editor Report · Acceptance letter]

PONE-D-25-00360R1

PLOS ONE

Dear Dr. Alvarez,

I'm pleased to inform you that your manuscript has been deemed suitable for publication in PLOS ONE. Congratulations! Your manuscript is now being handed over to our production team.

Kind regards,

on behalf of

Dr. Mohammad Irfan

Academic Editor

PLOS ONE